# Development of a Low-Frequency Piezoelectric Ultrasonic Transducer for Biological Tissue Sonication

**DOI:** 10.3390/s23073608

**Published:** 2023-03-30

**Authors:** Vytautas Ostasevicius, Vytautas Jurenas, Sandra Mikuckyte, Joris Vezys, Edgaras Stankevicius, Algimantas Bubulis, Mantas Venslauskas, Laura Kizauskiene

**Affiliations:** 1Institute of Mechatronics, Kaunas University of Technology, Studentu Street 56, LT-51424 Kaunas, Lithuania; 2Department of Mechanical Engineering, Kaunas University of Technology, Studentu Street 50, LT-51368 Kaunas, Lithuania; 3Institute of Physiology and Pharmacology, Lithuanian University of Health Sciences, A. Mickevicius Street 9, LT-44307 Kaunas, Lithuania; 4Department of Computer Sciences, Kaunas University of Technology, Studentu Street 50, LT-51368 Kaunas, Lithuania

**Keywords:** acoustic intensity, higher vibration mode, deep penetration, targeted acoustic wave, FEM

## Abstract

The safety of ultrasound exposure is very important for a patient’s well-being. High-frequency (1–10 MHz) ultrasound waves are highly absorbed by biological tissue and have limited therapeutic effects on internal organs. This article presents the results of the development and application of a low-frequency (20–100 kHz) ultrasonic transducer for sonication of biological tissues. Using the methodology of digital twins, consisting of virtual and physical twins, an ultrasonic transducer has been developed that emits a focused ultrasound signal that penetrates into deeper biological tissues. For this purpose, the ring-shaped end surface of this transducer is excited not only by the main longitudinal vibrational mode, which is typical of the flat end surface transducers used to date, but also by higher mode radial vibrations. The virtual twin simulation shows that the acoustic signal emitted by the ring-shaped transducer, which is excited by a higher vibrational mode, is concentrated into a narrower and more precise acoustic wave that penetrates deeper into the biological tissue and affects only the part of the body to be treated, but not the whole body.

## 1. Introduction

Medical ultrasound can be classified into diagnostic ultrasound and therapeutic ultrasound. A high-intensity focused ultrasound transducer could be used to necrotize a lesion deep within the human body. The authors of [1] present the development of a concave ring array transducer for medical therapy applications utilizing high-intensity focused ultrasound. The ultrasonic pressure distribution of the transducer was analyzed by deriving a theoretical equation, and a design scheme was proposed to optimize the transducer’s structure.

Different types of ultrasonic transducers are available depending on factors such as piezoelectric crystal arrangement, footprint, and frequency. A horn-shaped Langevin ultrasonic transducer was investigated in [2] to better understand the role of the acoustic profile in creating a stable trap. The characterization method included acoustic beam profiling via raster scanning with an ultrasonic microphone, as well as a finite element analysis of the horn and its interface with the surrounding air volume. A frequency domain solver was used to solve the linearized system of equations. The frequency of used in the study was 22.3 kHz, the same as in the experiment. The solver produced solutions for horn strain and stress, piezoelectric material stress and strain, and acoustic pressure distribution. The cup-shaped transducer has many applications in ultrasound technologies [3]. A cup-shaped ultrasonic transducer circuit was formed and the resonance/anti-resonance frequency equations were obtained. The vibration characteristics of the ultrasonic transducer were investigated by analytical and numerical methods and then confirmed by experiments. The results showed that the cup-shaped transducer possesses favorable vibrational properties: a large working mode amplitude, a uniform amplitude near the working surface, and a better isolation of the operating frequency from nearby untuned modes.

Ultrasound waves are mechanical waves that propagate through media. An article related to the propagation of ultrasonic waves in water [4] presents a simple method for generating a collimated ultrasound beam that exploits the natural Bessel-like vibration pattern of the radial modes of a piezoelectric disc with a side clamp. Both numerical and experimental studies were carried out to investigate the Bessel-type vibration patterns in the radial modes, showing excellent compatibility between the two studies. Ultrasonic beam profile measurements in water with a free and clamped piezoelectric transducer were presented. A collimated beam generation using lateral radial modes has significant applications for low-frequency imaging in highly attenuating materials. By mechanically constraining the lateral edges, the side-lobes have been significantly reduced, and a well-collimated beam has been generated. Experiments have been carried out to confirm the above findings, and it has indeed been found that the clamped transducer leads to a significant side-lobe suppression. In addition, the collimated beam of the clamped transducer had a greater penetration depth due to the absence of side-lobes. The paper [5] is a continuation of the previously discussed paper [4], in which numerical resonance and vibrational characteristics of the radial modes of a laterally stiff piezoelectric disc transducer are presented. The lateral stiffening was modeled using a spring, and the vibrational characteristics of the piezo-disc were investigated as the lateral stiffness increased from zero to a large value. The resonant frequency response was found to increase monotonically from free to clamped disc asymptotically. The role of lateral stiffness on ultrasonic propagation in water has been investigated by time-domain wave propagation studies.

The propagation of ultrasonic waves in an elastic body is investigated in [6]. The study discusses the characteristics of ultrasonic wave propagation in an isotropic elastic solid material due to the radial mode excitation of a piezoelectric disk actuator connected to its surface. Finite element simulation using coupled electromechanical modeling was used to study the wave propagation behavior. It was observed that radial mode vibrations at the surface of an elastic solid generated all three types of ultrasonic waves: longitudinal, shear, and surface waves. Solid-state waves consist of a central lobe and several side lobes based on the excitation frequency of the radial mode. The central lobe is primarily composed of longitudinal waves, while the side lobes are composed of shear waves. Furthermore, it was observed that longitudinal waves had fewer side lobes within the solid compared to shear waves. The results were obtained under the assumption of ideal bonding between the piezoelectric disc and the elastic solid, leading to a stiffness effect that reduced the side lobes of longitudinal waves, similar to the observations made in [4] for fluids. Materials with tailored acoustic properties are of great interest both for the development of tissue-mimicking phantoms for ultrasound research and smart scaffolds for ultrasound-based tissue engineering and regenerative medicine. The study presented in [7] evaluated the acoustic properties (sound velocity, acoustic impedance, and attenuation coefficient) of multiple materials with varying concentrations or cross-linking levels and barium titanate ceramic nanoparticle doping. The biological impact on human fibroblasts induced by low-intensity pulsed ultrasound-activated piezoelectric barium titanate nanoparticles was correlated with the precise ultrasound dose delivered. The findings indicate that proper acoustic characterization of the material enables accurate prediction of the ultrasound dose delivered to cells and the resultant bioeffects. Some specific mechanical and acoustic properties of human tissues have been observed to have minimal variability.

The aim of the paper [8] was to describe the development and validation of two low-intensity pulsed ultrasound stimulation systems capable of controlling the dose delivered to a biological target. A characterization of the transducer was carried out in terms of the shape and intensity of the pressure field in the high frequency range (500 kHz to 5 MHz) and for a low frequency value (38 kHz). This allowed the researchers to determine the distance along the beam axis where the biological samples should be located during stimulation and to know precisely the intensity at the target site. Time-domain acoustic modeling enabled accurate estimation of the ultrasound beam in the biological sample chamber, allowing precise control of the pressure delivered to the biological target by modulating the transducer input voltage. As demonstrated by acoustic simulations performed using the k-wave of the MATLAB acoustic toolbox, it was possible to have full control over the amplitude of the pressure acting on a target. Experiments simulating the conditions of future low-intensity pulsed ultrasound stimulation experiments showed that repeated immersion of the systems over a seven-day period did not alter the viability and metabolic activity of human primary chondrocytes. In addition, no macrophage activation was observed.

Paper [9] provides a concise overview of how biological cells behave when exposed to ultrasound only, i.e., without microbubbles. The phenomena are discussed from the physics and engineering perspectives. These phenomena include proliferation, translation, apoptosis, lysis, transient membrane permeation, and oscillations. The ultimate goal of cell acoustics is the detection, quantification, manipulation, and destruction of single cells. Detected differences in the rate of translation of individual cells could, in the future, serve as acoustic identifiers for cancer or malaria. Cell proliferation was enhanced by ultrasound at any frequency, implying that mechanical effects were not the likely cause of the observed proliferation. Hence, combined heating and acoustic vibration could be investigated as a means to accelerate the healing of injured tissue.

The authors in [10] investigate the biological effects of low-intensity ultrasound in vitro and review the factors that may enhance or inhibit these effects. The lowest possible ultrasound intensity required to kill cells or produce free radicals was determined. After sonication at this intensity, the effects of sonication in combination with hyperthermia, hypotonia, echocontrast agents, CO2, incubation time, high cell density, or various agents were investigated. The results showed that hyperthermia, hypotonia, and microbubbles are good enhancers of biological effects, while CO2, incubation time, and high cell density are good inhibitors. Cell membrane damage is a crucial factor in the events leading to cell death, and the mechanism of cell damage and repair is an important determinant of the fate of damaged cells.

The aim of the study in [11] was to develop a new ultrasound method based on the simultaneous observation of the change in ultrasound velocity and frequency spectrum of the signal propagating in coagulating blood, and to apply it to the automatic estimation of blood coagulation parameters. The results have shown that the ultrasound velocity and the frequency spectrum of the ultrasound signal should be used simultaneously during blood clotting to determine the onset and duration of clot retraction. The results confirmed that clot retraction was influenced by fibrinogen concentration and platelet receptor activity, which are determined by carrier genotype.

As a result of COVID-19, the need for and variety of pulmonary therapy devices has increased. The aim of the study [12] was to evaluate whether low-frequency ultrasound can be used to detect air trapping in chronic obstructive pulmonary disease. In addition, the ability of low-frequency ultrasound to detect the effects of short-acting bronchodilators was evaluated. Ultrasound at a frequency of 20–40 kHz was transmitted to the sternum and received in the back during inspiration and expiration. The high pass rate was determined from the inspiratory and expiratory signals and their difference. A significant difference in inspiratory and expiratory signals was found between subjects with chronic obstructive pulmonary disease and healthy subjects. It was concluded that low-frequency ultrasound is cost-effective, easy to perform, and suitable for detecting air trapping.

In [13], to assess the effect of positive end-expiratory pressure on ultrasound propagation through injured lungs, eight anaesthetized, intubated, and mechanically ventilated pigs were injected with multifrequency broadband sound signals into their airways, and the transmitted sound was recorded at three locations bilaterally on the chest wall. Oleic acid injections caused severe pulmonary edema, mainly in the dependent lung regions, where a concomitant decrease in sound transmission time was observed (*p* < 0.05), while no statistically significant changes occurred in the lateral or non-independent regions. Positive end-expiratory pressure resulted in a reduction in venous impaction, an increase in respiratory compliance, and a return of sound transmission time to pre-injury levels in the dependent lung regions.

A human ultrasound [14] study was conducted to evaluate the effect of acoustic signal transmission utilizing a pair of transducer detectors, or a 12-sensor elastic chest belt, positioned 5 cm apart and wrapped around the thorax, and a single pulse transmitter attached to the sternum. The assessment did not include an analysis of the echoes. The transmission of ultrasound through the thorax and lungs between 1 Hz and 1 MHz was found to exhibit three distinct frequency bands: an acoustic signal < 1 kHz was transmitted at a velocity of 30–50 m/s, no transmission was recorded between 1 and 10 kHz, and ultrasound with frequencies > 10 kHz was transmitted at a speed of 1500 m/s. It was demonstrated that low-frequency ultrasound (10–750 kHz) can penetrate the thorax and provide information on air and fluid content within human lungs. The transmission of expiratory acoustic signals was significantly reduced in patients with pulmonary emphysema or pneumothorax, but increased in those with pleural effusions. It was concluded that low-frequency ultrasound transmitted through the lungs can be applied as a non-invasive real-time diagnostic method.

The study [15] examines the acoustic behavior of flexibly coated microbubbles and rigid coated microcapsules and their contribution to improved drug delivery. Bubble vibration is dictated by acoustic ultrasound parameters such as frequency, pulse length, amplitude, and repetition rate and induces hydrodynamic effects around the oscillating microbubbles. The theoretical and experimental evaluation of phenomena related to drug delivery, such as non-spherical oscillations, shear stress, microstreaming, and atomization, is conducted in relation to two drug delivery systems, co-administration and microbubble-based drug carriers. Mechanical coupling has received limited investigation due to the small vibrational timescale of microbubbles (nanosecond to microsecond), which is significantly shorter than the timescale of physiological (millisecond), biological (seconds to minutes), and clinical (days to months) effects. Furthermore, cell death and drug delivery can only be monitored indirectly, usually using fluorescent staining, which requires specialized equipment.

The study presented in [16] systematically examined the transient enhancement of cell membrane permeability in endothelial cells and in two breast cancer cell lines. The use of ultrasound in combination with microbubbles was demonstrated to facilitate the internalization of drugs into cells. The duration of the transient increase in cell membrane permeability after ultrasound exposure varied between 1 and 3 h among the different cell lines.

The biophysical effects of ultrasound, including thermal and non-thermal effects on cells, have been investigated by the authors in [17]. The results showed that ultrasound irradiation can increase the permeability of cell membranes due to the sonophore effect, allowing molecules such as drugs, proteins, and DNA to pass through cell membranes. Optimal parameters have been determined to enhance the therapeutic efficiency of the chemotherapeutic drug MDA-MB-231.

The literature review concluded that there is no evidence of the ability to excite higher ultrasound transducer oscillation modes, which would not only increase the penetration or acoustic pressure of the ultrasound acoustic signal, but would also allow for a more precise targeting of therapeutically affected tissues.

## 2. Materials and Methods

This paper presents the results of a study on digital twins. The term “digital twin” still lacks a common understanding, leading to differences in its technological implementation and objectives. This term covers virtual and physical replicas of a device under development, which are used as a specific test-bed for a process or a product to simulate the changes made before they are implemented in real life by entitling virtual and physical copies as virtual and physical twins, respectively, and linking them to simulations and experiments, resulting in a digital output. Acoustic waves carry energy that can be harnessed to perform useful work. The energy density carried by a plane wave is given by [18]
*ε* = *P v*/2 *c* = *P*^2^/2 *ρ c*^2^ = *ρ v*^2^/2(1)
where *P* and *v* are the acoustic pressure and velocity amplitudes, *ρ*_0_ represents the mass density, and *c* is the sound velocity of the medium.

Another commonly used metric for describing energy propagation in a wave is the wave intensity, which quantifies the rate of energy transfer by the acoustic wave (in units of Wcm^−2^). The time average intensity for a plane wave in a fluid can be calculated directly from the pressure and fluid properties. The sound intensity *I* and the sound pressure *P* are two characteristic parameters describing the acoustic wave propagation and are defined by the following equation [18]:*I* = *P*^2^/2*ρ*_0_*c* = 0.5 *ρ*_0_*cω*^2^*A*^2^(2)
where *ρ*_0_ represents the mass density and *c* is the sound velocity of the medium (1 g/cm^3^), *ω* is the angular frequency, and *A* is the amplitude of the acoustic wave.

An additional important material property for the design of acoustic systems is the material attenuation coefficient. Attenuation describes the irreversible loss of acoustic energy to heat due to various mechanisms such as viscosity or molecular relaxation [18]. When an acoustic wave propagates in a material, the pressure amplitude after a distance *L* is given by
*P* = *P*_0_ e^−αL^,(3)
where *P_0_* was the initial pressure of the wave and α is the attenuation coefficient in neper per centimeter. Attenuation is highly frequency-dependent, with higher frequencies being attenuated more strongly than lower ones. Thus, at frequencies below 0.3 MHz, ultrasound has a significantly better effect on deeper biological tissues, and a strong biochemical reaction is more likely to occur. The effects of low-frequency ultrasound on bones, blood vessels, and internal organs should therefore be carefully studied. However, cavitation phenomena below 100 kHz can destroy biological tissues and, in some places, raise their temperature above the vital limit. The mechanical index (MI) is an indication of the mechanical damage that may be caused by inertial cavitation:*MI* = *P_NP_/f_c_*^−0.5^(4)
where peak-negative pressure (*P_NP_*) is expressed in MPa and *f_c_* is expressed in MHz. The value taken for *P_NP_* should be the maximum value anywhere in the field, measured in water but reduced by 0.3 dB cm^−1^ MHz^−1^ attenuation.

Furthermore, in this ultrasound frequency range (20–100 kHz), emulsification or dispersion forces acting on blood can easily cause hemolysis. These factors are also frequency- and intensity-dependent. Therefore, the effects of ultrasound on bones, blood vessels, and internal organs must be carefully studied when developing new types of ultrasound emitters operating below 100 kHz.

Three different configurations of Langevin-type ultrasonic transducers were developed. One of them had a cylindrical front mass with a flat ultrasound emitting surface (Figure 1a), and the other two had different diameters d_1_ of 59 mm and of 100 mm, respectively, with a ring-shaped front mass surface (Figure 1b). The design of the latter is based on three concepts: (1) the use of a front mass with a ring-shaped front mass surface produces stronger excitation; (2) the radial mode oscillation of the modified front mass produces a concentrated acoustic field; and (3) more acoustic energy is produced in the higher-frequency vibrational mode.

The fabricated transducers were composed of two piezo-ceramic rings (material—PZT-4), a steel cylinder-shaped back mass (St 304), and an aluminum cylinder-shaped front mass (Al 7075-T6).

### 2.1. Ultrasonic Transducer Virtual Twin

The FEM models of three Langevin-type ultrasonic transducers with different front mass designs were investigated as virtual twins by comparing two Langevin-type ultrasonic transducers with different diameters of 58 mm and 100 mm with a ring-shaped surface and one with a flat front mass surface. Comsol Multiphysics 5.6 software was used to create the three-dimensional finite element models (FEM) of the piezo transducers and to perform simulations. This software was used to build a full 3D FEM model and to analyze the transducers in order to observe their vibration behavior through simulation by modal analysis and to determine their natural frequencies by harmonic analysis. This was also carried out in order to establish the validity of the analytical results. The piezoelectric transducers were modeled using a 3D modeling approach, and mesh elements were used for piezoelectric and other components. Modal analysis was used to determine the natural frequencies, mode shapes, and the location of the nodal plane. This analysis was performed under resonance conditions with a constant voltage of 50 V applied to the electrical contacts of both ceramic disks. No structural constraint was applied to the modal analysis. This simulates an unconstrained transducer assembly. This state is similar to the physical test state where the transducer is without any constraints. The properties of the materials used for modeling are listed in two tables: Table 1 and Table 2.

Muscle material properties (Table 3) were selected for the analysis of acoustic wave propagation in the human tissue. The attenuation coefficient of a material is considered to be frequency-dependent. The form *α* = *α*_0_ ∙ *f^b^* is assumed, where *α* (Np/m) is the absorption coefficient for a given frequency *f*, *α*_0_ (Np/m/Hz) is a medium constant, and *b* is also a numerical constant dependent on the tissue type [22].

FEM modeling was used to investigate the vibration modes and sound pressure field in the range of 0–100 kHz, including the radiation in the muscle at the two lowest resonant modes of the developed piezoelectric transducer.

### 2.2. Ultrasonic Transducer Physical Twin

A piezoelectric transducer with a flat front mass was purchased as ultrasonic cleaning generator driver board + 60 W 28 KHz Transducer (OKS Ultrasonic Group Co., Ltd., Beijing, China), and a modified ring-shaped 58 mm diameter transducer has been manufactured in our university laboratory from a similar purchased transducer.

The dynamics of the transducers were evaluated by measuring the electrical impedance, resonant frequencies, and vibration modes and comparing them with the simulation results. A Polytec Laser Doppler 3D scanning vibrometer PSV-500-3D-HV (Polytec GmbH, Waldbronn, Germany) and a linear amplifier P200 (FLC Electronics AB, Partille, Sweden) were used for a high-precision measurement of the three-dimensional vibration distribution on the front surface of the transducer (Figure 2).

With the help of Laser Doppler Velocimetry (LDV), it was possible to determine not only the resonance frequencies of the vibrations of the surface radiating acoustic energy of the front mass, but also the mode of vibrations of the separated points. This revealed whether the transducer was excited by a longitudinal or a radial mode of vibrations. Additionally, an impedance analyzer 6500 B (Wayne Kerr Electronics Ltd., Bognor Regis, UK), presented in Figure 3, was used to measure the frequency dependence of the transducers under investigation. The obtained impedance graphs show not only the resonances of the longitudinal and radial vibrations of both transducers that coincide with those measured by LDV, but also the resonances of other elements that make up the transducer.

## 3. Simulation and Experimentation Results

Numerical FEM simulations of the transducers were performed, and the vibration modes and resonant frequencies were determined using forced harmonic analysis. Three Langevin-type ultrasonic transducers with different front mass designs were investigated: two of them with diameters d_1_ of 58 mm and 100 mm and ring-shaped front mass surfaces and one with a flat front mass surface. The amplitude–frequency characteristics of the transducers with flat and ring-shaped front mass surfaces are presented in Figure 4 in the *X*, *Y*, and *Z* axis.

The modal shapes of the piezoelectric transducer with flat and cut-out surfaces at the first natural frequency vibrations are shown in Figure 5.

The modal shapes of the piezoelectric transducers with flat and ring-shaped front surfaces at the second natural frequency vibrations are presented in Figure 6.

The propagation of acoustic waves in muscle excited by piezoelectric transducers with flat and cut-out surfaces in the first mode are presented in Figure 7.

The propagation of acoustic waves excited by the flat and ring-shaped surfaces of the piezoelectric transducers in muscle at the second mode are presented in Figure 8.

The distribution of the acoustic pressure level of an ultrasound wave propagating in a muscle tissue, simulated by piezoelectric transducers, in the second natural mode is presented in Figure 9.

The dynamics of the two fabricated transducers with different front mass designs of 58 mm have been evaluated by measuring the electrical impedance, resonant frequencies, and vibration modes and comparing them with the simulation results (Figure 10a,b).

A Polytec Laser Doppler 3D scanner was used for high-precision measurement of the three-dimensional vibration distribution of the transducer’s front surface (Figure 2). A periodic chirp type driving signal of 50 V was used in the frequency range from 20 kHz to 100 kHz (Figure 11a,b). The longitudinal and radial vibrational modes with the highest velocity amplitudes were measured with a Polytec 3D scanning vibrometer at two resonance frequencies: a longitudinal amplitude of 12.8 mm/s at 28.47 kHz and a radial amplitude of 25.5 mm/s at 46.19 kHz for the transducer with a flat surface, and a longitudinal amplitude of 9.5 mm/s at 28.13 kHz and a radial amplitude of 42.7 mm/s at 38.04 kHz for the transducer with a ring-shaped surface. A stronger excitation on the second natural frequency was obtained by utilizing a transducer with a ring-shaped front mass surface.

Comparing with the two resonant frequencies of the piezoelectric transducers, measured with an impedance analyzer, for the transducer with a flat surface are 29 kHz and 46 kHz. For the transducer with ring-shaped surface, the frequencies are 28 kHz and 39 kHz and they coincide with frequencies of the first and second resonant modes, as measured with a Polytec 3D scanning vibrometer. Since the Langeven-type piezoelectric transducer is composed of five structural elements (two piezoceramic disks, a screw, front and rear masses), which have their own resonant frequencies, the impedance curve shows significantly more “peaks”, compared to the curve obtained by LDV, which shows only the frequency dependence of the vibration level of the surface of the front mass which generates the acoustic wave.

Since the ethical permission for experimental biological studies has not yet been obtained, the determination of the acoustic characteristics of the ultrasound intensity of the created piezoelectric transducers took place in a water bath (dimensions—240 × 140 × 100 mm) with a hydrophone HCT-0320 connected to an acoustic cavitation meter MCT-2000 (Onda Corp., Sunnyvale, CA, USA). The ultrasound intensity was measured at longitudinal and radial first-order resonant mode frequencies obtained with a Polytec 3D scanning vibrometer. At 120 V, the RMS values of the ultrasound intensity in the longitudinal/radial vibration modes were: 200/120 mW/cm^2^ for the transducer with a flat surface and 130/85 mW/cm^2^ for the transducer with a ring-shaped surface. The distance between the output surface of the transducer and the hydrophone was kept at about 70 +/− 10 mm.

## 4. Discussion

The Langevin-type ultrasonic wave transducer has been analyzed for its high directivity and long propagation distance properties due to its high frequency (>25 kHz) and short wavelength, and it has been extensively studied for detection and sensing purposes. Ultrasound sonication is known to have effects on the living body, such as the promotion of enzyme reactions, emulsification, thermogenic effects, expansion of capillary blood vessels, and improving metabolism. Here, “the effective depth” of ultrasound exposure is defined as the depth of the ultrasound beam close to the body surface of the patient’s internal organ, which is effectively treated therapeutically. In this case, the acoustic energy delivered to the organ is proportional to the intensity and duration of the ultrasound. The first five resonant frequencies of the developed 58 mm diameter transducers were modelled, and further analysis showed that the vibrational modes with the largest displacements were found at two resonant frequencies: around 29 kHz and 46 kHz for the flat surface transducer, and around 29 kHz and 40 kHz for the ring-shaped front mass transducer (Figure 4a,b). The resonant frequencies of the developed 58 mm diameter piezoelectric transducers measured with an impedance analyzer (Figure 10) coincide with the frequencies of the first and second resonant modes, measured with a Polytec 3D scanning vibrometer (Figure 11), and the resonant frequencies determined using FEM modeling (Figure 4a,b).

The simulated vibrational modes of the piezo transducer at the first natural frequency indicate that the transducer with a flat surface vibrates only in the *Z* direction of the longitudinal axis, while the ring-shaped surface transducer is excited by longitudinal and radial vibrations (Figure 4). In the case of a cut-out surface type transducer, since the tip volume is cut out, more deformation is observed in the *X*- and *Y*-axes. The propagation of acoustic waves excited by flat and ring-shaped surface piezoelectric transducers in muscle tissue at the second mode show that the radial vibrations are dominating for the transducers with flat and ring-shaped surfaces, and amplitudes of vibration in the *X*–*Y* directions are higher for transducers with cut-out surfaces (Figure 9). The acoustic pressure wave varies above and below the ambient pressure, typically with harmonic (sinusoidal) modulation. Free-field conditions are used for measurements of acoustic pressure. These conditions are approximate to those under which the acoustic field consists only of a traveling wave, propagating into an infinite medium without boundaries. Due to the relatively thin wall, this type of transducer is able to generate a more directional and concentrated acoustic wave, as can be seen in Figure 9b. Half of the sphere, 200 mm in radius, was used for acoustic wave modelling in a muscle medium. A perfectly aligned area of 10 mm layers was also used to simulate an open and non-reflecting infinite region in order to match the model to realistic conditions.

As illustrated by the graphs (Figure 9a), the total acoustic pressure of the ring-shaped surface transducer type is lower than that of the flat-surface transducer type, but it is more directional, which is very important when trying to use it in a real-world environment. However, the same Figure 9b shows that the acoustic pressure generated by a 100 mm ring-shaped transducer at a 6 cm depth in the muscle tissue is almost five times higher than the acoustic pressure of a 58 mm diameter ring-shaped surface transducer. When used to treat the human body, the more directional and precise the wave, the better the results that can be achieved, as only the part of the body being treated is affected, not the whole body. Furthermore, the longitudinal mode is not as suitable for medical applications as the radial mode because the stronger acoustic signal produced by the second natural frequency mode transducer can be focused on a specific depth of biological tissue. Asymmetry between positive and negative half-cycles can be seen for the simulations of acoustic pressure level distribution in the muscle medium, and it is caused by the non-linear propagation of the ultrasound wave in an interface layer between the transducer and the muscle tissue. Under such conditions, the peak rarefaction, *p_r_*, and peak compression, *p_c_,* or peak negative and peak positive pressure, respectively, are separately identified. In addition, only the peak rarefaction is used to estimate the risk of the sonicated tissue destruction due to the mechanical cavitation. In this case, a Mechanical Index (*MI*) is used, which is given by the following equation: *MI = p_r_ f*^0.5^, wherein *p_r_* is the maximum negative peak pressure in units of MPa, and *f* is the center frequency of the ultrasonic wave in units of kHz. Thus, it is a frequency-weighted acoustic pressure value, and it indicates the likelihood of cavitation when the *MI* is higher than 0.6. Therefore, the very high peak negative acoustic pressure generated by the flat surface transducer in an interface layer between the transducer and muscle tissue (Figure 9a) can induce tissue destruction by cavitation.

A low-frequency ultrasound can be used to stimulate blood flow in patients with pulmonary hypertension, to improve the susceptibility of biofilm-associated S. aureus to antibiotics on the surface of prostheses, and to treat endothelium dysfunction, by facilitating the effects of drugs on liver and kidney cells to identify clinically relevant biomarkers related to the function of those cells.

## 5. Conclusions

By applying the methodology of digital twins, a new type of low-frequency ultrasound transducer was designed, manufactured, and tested for deep human tissue therapy. The ability of the proposed transducers to penetrate into deeper human tissues is associated with the excitation of the transducer’s natural vibrations in higher modes. To prove this, the proposed transducer was modeled, and the simulation results were experimentally validated. The excited higher vibrational mode of the transducer increased the penetration of the acoustic signal, and the signal became less scattered, which made it possible to increase the acoustic effect in deeper biological tissues and apply the developed device to therapeutic applications.

## Figures and Tables

**Figure 1 sensors-23-03608-f001:**
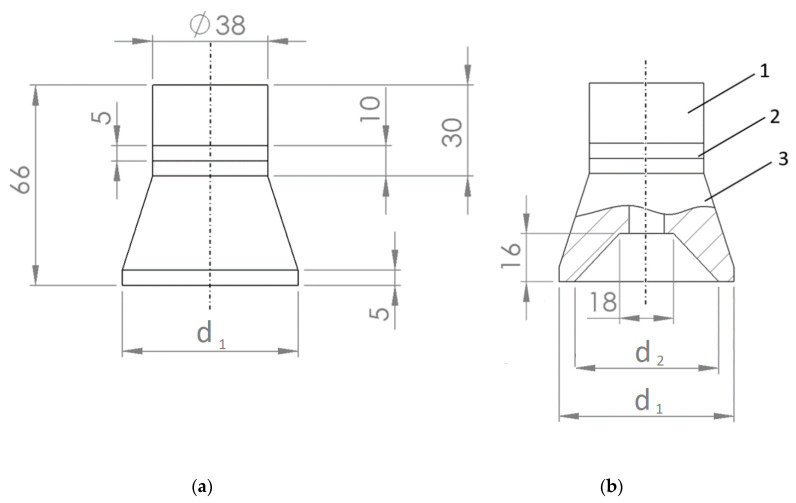
Geometric dimensions of the piezo transducer in mm with flat surface (**a**) and ring-shaped surface (**b**) of the front mass 3: 1—back mass; 2—two ring-shaped piezoelectric elements; 3—front mass.

**Figure 2 sensors-23-03608-f002:**
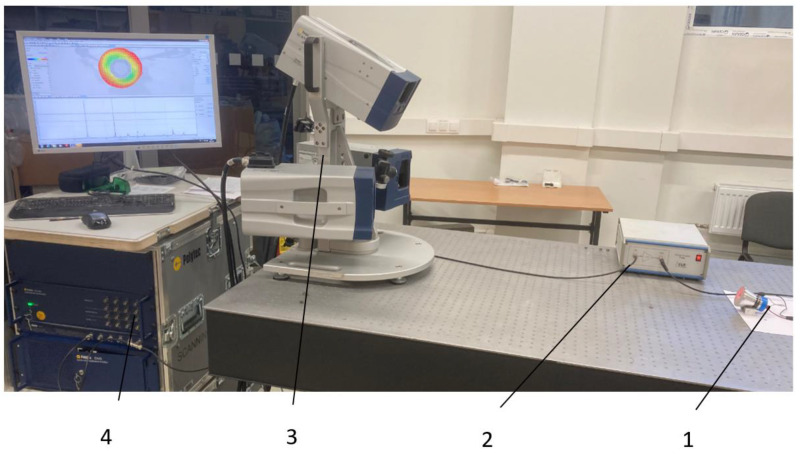
Set-up with Polytec PSV-500-3D scanning laser vibrometer: 1—transducer; 2—linear amplifier; 3—Polytec scanning laser head; 4—Polytec signal generator/data acquisition system.

**Figure 3 sensors-23-03608-f003:**
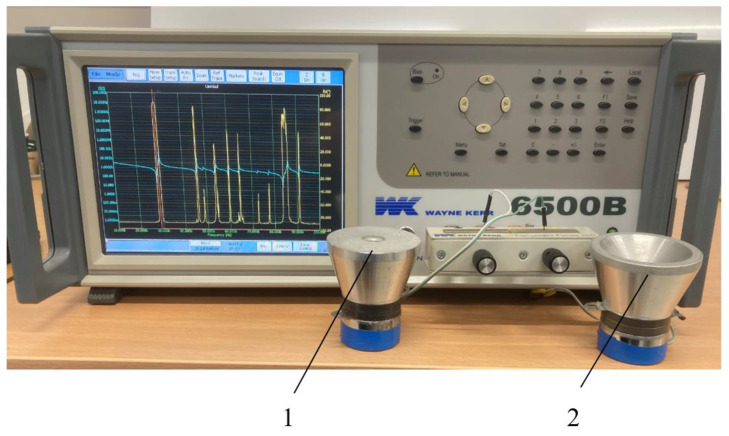
Experimental setup with impedance analyzer 6500 B (Wayne Kerr Electronics Ltd., Bognor Regis, UK) and the two tested transducers: 1—transducer with flat surface; 2—transducer with ring-shaped surface.

**Figure 4 sensors-23-03608-f004:**
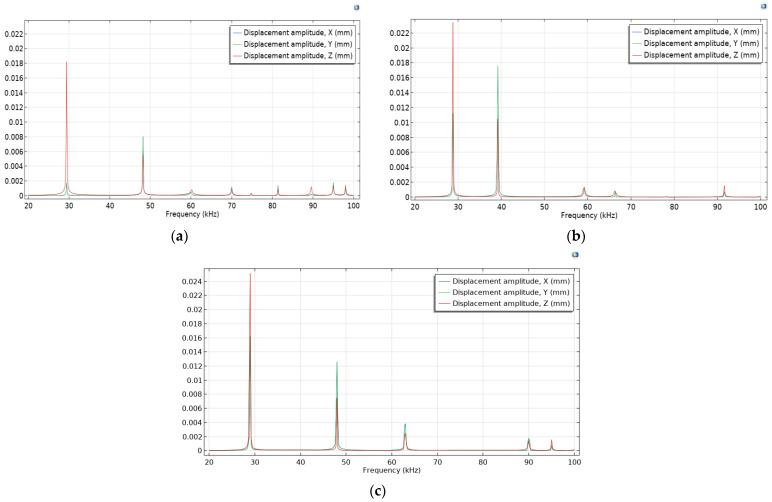
Theoretical amplitude–frequency characteristics of piezoelectric transducers using Comsol multiphysics: (**a**) with flat surface; (**b**) with ring-shaped surface with 59 mm outer diameter; (**c**) with ring-shaped surface with 100 mm outer diameter.

**Figure 5 sensors-23-03608-f005:**
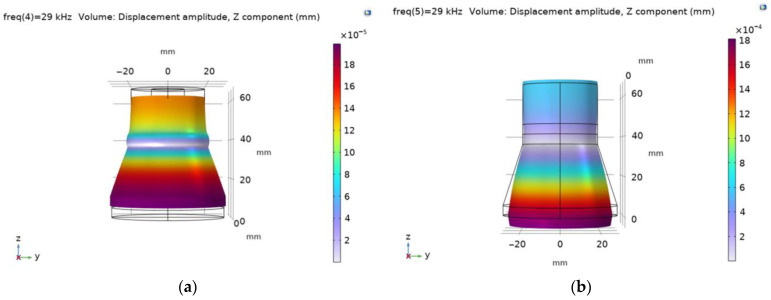
Modal shapes of the piezoelectric transducer at the first natural frequency: (**a**) with flat surface; (**b**) with 58 mm diameter ring-shaped surface; (**c**) with 100 mm diameter ring-shaped surface.

**Figure 6 sensors-23-03608-f006:**
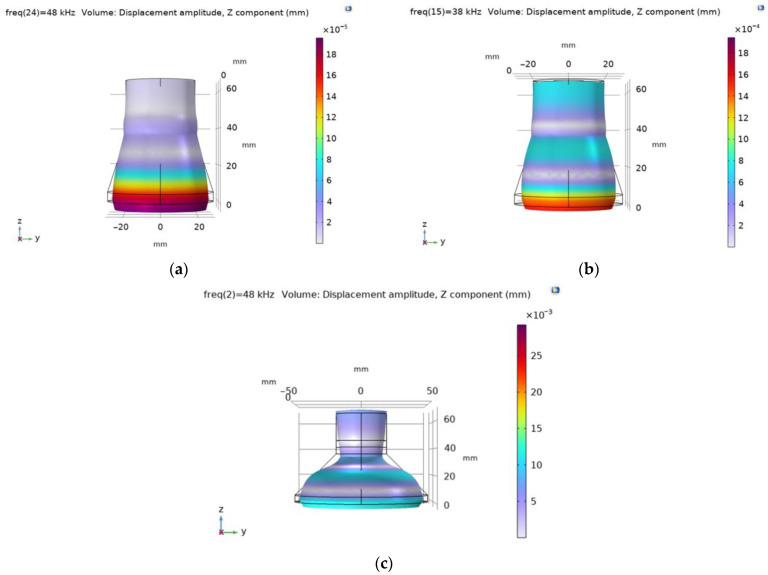
Modal shapes of the piezoelectric transducer at the second natural frequency vibrations: (**a**) with flat surface; (**b**) with 58 mm diameter ring-shaped surface; (**c**) with 100 mm diameter ring-shaped surface.

**Figure 7 sensors-23-03608-f007:**
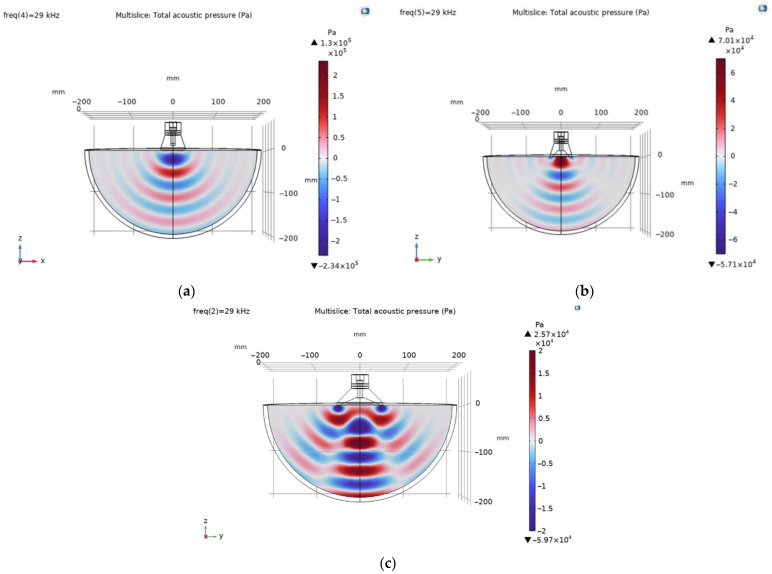
Propagation of an acoustic field generated by piezoelectric transducer at the first mode in the muscle: (**a**) with flat surface; (**b**) with 58 mm diameter ring-shaped surface; (**c**) with 100 mm diameter ring-shaped surface.

**Figure 8 sensors-23-03608-f008:**
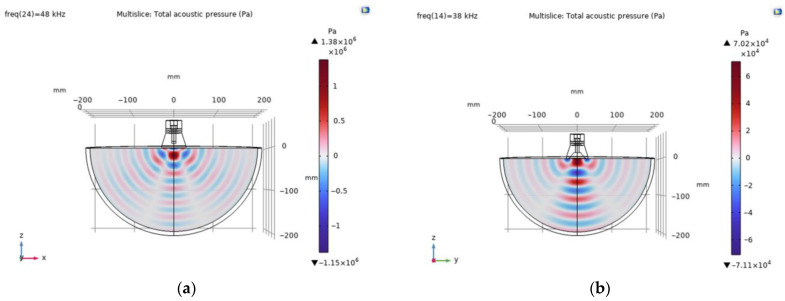
Propagation of acoustic field generated by piezoelectric transducer at the second mode in the muscle: (**a**) with flat surface, (**b**) with 58 mm diameter ring-shaped surface, (**c**) with 100 mm diameter ring-shaped surface.

**Figure 9 sensors-23-03608-f009:**
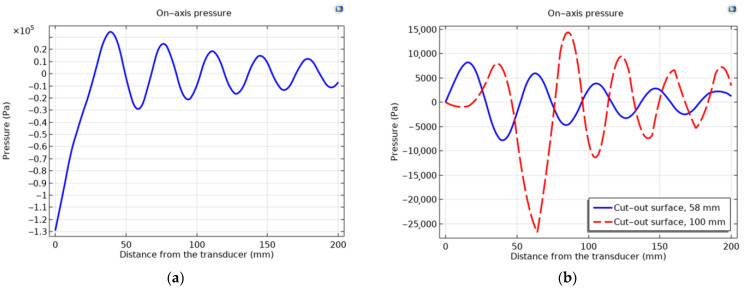
Distribution of the acoustic pressure level of an ultrasound wave propagating in a muscle medium, simulated by piezoelectric transducers, in the second natural mode: (**a**) with a flat surface at a resonant frequency of 48 kHz; (**b**) with a ring-shaped surface and the different diameters of: 58 mm at a resonant frequency of 38 kHz (solid line) and with a diameter of 100 mm at resonant frequency of 47 kHz (dashed line).

**Figure 10 sensors-23-03608-f010:**
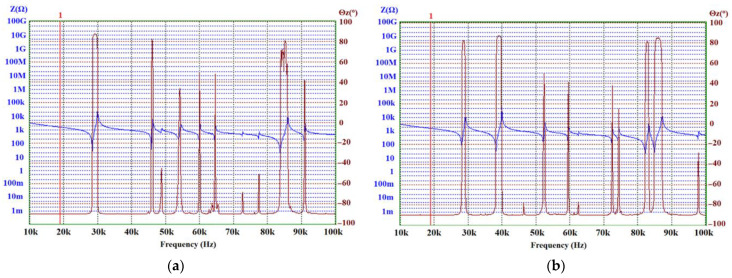
Experimental impedance–phase vs. frequency characteristics: (**a**) the electric impedance of the transducer with flat surface; (**b**) the electric impedance of the transducer with ring-shaped surface.

**Figure 11 sensors-23-03608-f011:**
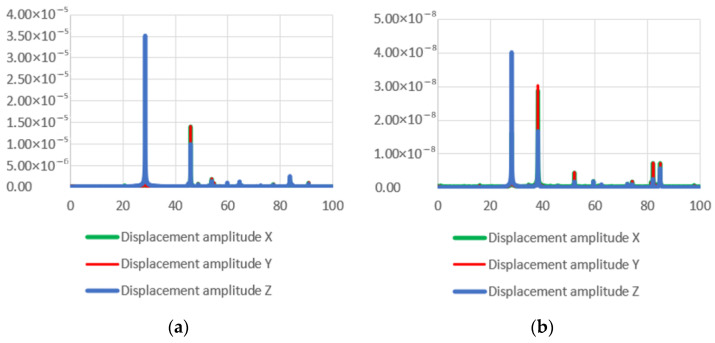
Experimental amplitude–frequency characteristics of the two developed transducers measured with Polytec 3D scanning vibrometer: (**a**) with flat surface; (**b**) with ring-shaped surface.

**Table 1 sensors-23-03608-t001:** Material properties of PZT-4.

Material Properties	Piezoceramic PZT-4 [19]
Density [kg/m^3^]	7500
Dielectric permittivity matrix, ×10^−7^ [F/m]	ε_11_ = 11.42; ε_33_ = 8.85
Piezoelectric matrix [C/m^2^]	e_13_ = −18.01; e_33_ = 29.48; e_52_ = 10.34
Elasticity matrix, ×10^10^ [N/m^2^]	c_11_ = 14.68; c_12_ = 8.108; c_13_ = 8.105; c_33_ = 13.17; c_44_ = 3.29; c_66_ = 3.14

**Table 2 sensors-23-03608-t002:** Properties of the materials used for simulation.

Material Properties	Steel [20]	Aluminum [21]
Young’s modulus [N/m^2^]	200 × 10^9^	71.7 × 10^9^
Poison’s ratio	0.25	0.33
Density [kg/m^3^]	7800	2810

**Table 3 sensors-23-03608-t003:** Material properties of muscle [22].

Speed of sound, m/s	1588
Density, kg/m^3^	1090
Attenuation coefficient Np/m	0.206 (38 kHz); 0.265 (48 kHz)

## Data Availability

Data sharing not applicable.

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
