# Peer review of "Development of a Low-Frequency Piezoelectric Ultrasonic Transducer for Biological Tissue Sonication"

_sensors, 2023, doi:10.3390/s23073608_

Round 1
Reviewer 1 Report
This paper presents the development of a low-frequency piezoelectric transducer that works at low frequencies (20 to 100 kHz). Though the title suggests that the transducer is developed for biological tissue sonication, no experimental details or results are included in this paper in this regard.
Specific comments:
1. The abstract starts with the intensity of the ultrasound levels recommended for its safe use in human subjects, but the paper doesn’t include any details on the measurement of the ultrasound intensity of the fabricated transducer. The paper only describes impedance measurements and LDV characterization. The paper will add value only when the points mentioned in the abstract are adequately described in the body of the paper. Therefore, acoustic characterization of the transducer must be included in the revised version of the paper.
2. The abstract should reflect the content of the paper. Here, the abstract has no relation to the content reported in the paper.
3. The content of the paper is not organized carefully. For example, figure 2 shows the LDV characterization technique, but no explanation of the same. The next session deals with COMSOL simulation, later then the details of the LDV characterization are given. A paper should have a good flow for the reader to go through. Please rework the same.
4. Figure 10(a) shows several peaks in the impedance data, which are not really harmonics of the fundamental frequency. From where do those peaks appear?
5. It would be good to compare the LDV frequency response with the impedance characteristics. That will give an idea of the peaks in the impedance characteristics.
6. The paper should be thoroughly reworked in terms of grammar and way of presentation.
Author Response
We appreciate the time and effort on the part of the editor and referees in reviewing this manuscript and providing constructive comments that helped to improve and clarify the manuscript considerably. We hope that the revised version of the manuscript answers their concerns.

Reviewer 2 Report
In this manuscript, the authors proposed a new type of low-frequency ultrasound transducers for deep human tissue therapy by applying the methodology of digital twins. The main contribution and motivation are that the excited higher vibrational mode of the transducer increased the penetration of the acoustic signal, the signal became less scattered, which made it possible to increase the acoustic effect in deeper biological tissues and apply the developed device to therapeutic applications.
The points below will help improve the manuscript.
1. In Table 1., the Young’s modulus of PZT-4 should be clarified, and since the PZT-4 is generally transversely isotropic material, its longitudinal and transverse Young's modulus are different.
2. Figure 3 does not add any value -- please consider omitting.
3. As author mentioned “the proposed transducer was modeled and the simulation results were experimentally validated.” in section 5, can you add the results of measuring the actual acoustic field generated by your transducers with a hydrophone, at least, please add the experimental results of sound pressure versus the distances from the ultrasonic transducer.
4. Materials section, where was the piezoelectric transducer purchased/manufactured. If the researchers manufactured it, which method did they use? And how did they ensure that it operates at the correct frequency?
5. A minor note, it is difficult to distinguish the three curves in Fig. 11.
6. Can you discuss the specific potential therapeutic applications with the transducer?
Author Response

(The authors gave the same response as above.)

Round 2
Reviewer 1 Report
This paper may be accepted for publication in the current form.